# Quantifying confidence shifts in a BERT-based question answering system evaluated on perturbed instances

Ke Shen, Mayank Kejriwal *

Information Sciences Institute, University of Southern California, Marina del Rey, California, United States of America

☯ These authors contributed equally to this work.
* kejriwal@isi.edu

**Data Availability Statement:** All files and data used in this study are available from the Kaggle platform (doi: 10.34740/kaggle/dsv/5130107).

## Abstract

Recent work on transformer-based neural networks has led to impressive advances on multiple-choice natural language processing (NLP) problems, such as Question Answering (QA) and abductive reasoning. Despite these advances, there is limited work still on systematically evaluating such models in *ambiguous* situations where (for example) no correct answer exists for a given prompt among the provided set of choices. Such ambiguous situations are not infrequent in real world applications. We design and conduct an experimental study of this phenomenon using three probes that aim to 'confuse' the model by perturbing QA instances in a consistent and well-defined manner. Using a detailed set of results based on an established transformer-based multiple-choice QA system on two established benchmark datasets, we show that the model's confidence in its results is very different from that of an expected model that is 'agnostic' to all choices that are incorrect. Our results suggest that high performance on idealized QA instances should not be used to infer or extrapolate similarly high performance on more ambiguous instances. Auxiliary results suggest that the model may not be able to distinguish between these two situations with sufficient certainty. Stronger testing protocols and benchmarking may hence be necessary before such models are deployed in front-facing systems or ambiguous decision making with significant human impact.

## Introduction

Question Answering (QA) [1] and inference are important tasks in natural language processing (NLP) and applied AI, and fundamental to the development of conversational 'chatbot' agents [2]. Advancements in the last five years in deep neural transformer-based models have led to significant improvements in QA performance, especially in the multiple-choice setting. Bidirectional Encoder Representations from Transformers (BERT) [3] was a pivotal model that ushered in a wave of transformer-based architectures that consequently achieved state-of-the-art performance in a range of NLP tasks, including QA and Web search. BERT achieved state-of-the-art performance due to its self-attention mechanism, novel bidirectional encoding capability, masked language modeling, and next sentence prediction functions.

**Funding:** MK received funding for this work as a principal investigator of MOWGLI, a project in the Defense Advanced Research Projects Agency (DARPA) Machine Common Sense program, supported by the United States Office Of Naval Research under Contract No. N660011924033. The funders had no role in study design, data collection and analysis, decision to publish, or preparation of the manuscript.

**Competing interests:** The authors have declared that no competing interests exist.

While generative models such as ChatGPT and GPT-3 have recently captured much of the general public's attention [4], BERT-based models continue to be important in many applications, especially those that rely on fine-tuned domain-specific data (often of a proprietary or specialized nature) and open-source architecture that can be efficiently executed on private servers. Examples of such specialized BERT-based models include Patentbert [5], Docbert [6], SciBERT [7], DistilBERT [8] and K-bert [9], all of which have achieved groundbreaking results in diverse language understanding tasks, including QA [10–12], text summarization [13, 14], sentence prediction [15, 16], dialogue response generation [17, 18], natural language inference [19, 20], and sentiment classification [21–23]. Because BERT-based models, and their many applications, continue to be important in real-world settings, understanding its performance in unexpected and ambiguous situations has also become important, especially in broad domains such as commonsense multiple-choice QA.

Although a growing body of research continues to be published on so-called BERTology [24], which aims to understand the psycholinguistic properties, and knowledge content, of models like BERT, there is considerable less work on systematically evaluating such models in *ambiguous* situations where (for example) no correct answer exists for a given prompt among the provided set of choices. Such ambiguous situations are not infrequent in real world applications [25].

Given real-world usage and relevance of language representation models like BERT, this article aims to systematically study their behavior in such ambiguous situations. We design and implement a novel set of three *confusion probes* to observe how the confidence distribution shifts, when faced with ambiguity, of an established multiple-choice QA system based on the RoBERTa model, an evolution of BERT. Derived from the original BERT model, RoBERTa modified its pre-training objectives, adopted larger batch sizes and longer sequences, and used dynamic masking techniques during training. These enhancements have demonstrated RoBERTa's robustness and its ability to achieve superior performance across various NLP tasks when compared to BERT. The confusion probes, which are conceptually simple to design and hence replicate, are illustrated in Fig 1, which also introduces some important terminology that we subsequently define in the *Formalism* section. Each probe operates at the level of an *instance*, which comprises a *prompt* and a *candidate set of answers* or choices. While a prompt can be a question, in practice, it tends to be more complex, and may be a statement, paragraph, or even a contextualized question, as demonstrated in Fig 1. Based on the specific commonsense task that the QA benchmark is seeking to evaluate, exactly one of the candidate choices is considered correct for that prompt. However, once a confusion probe is applied, there is no correct choice. Probes may perturb the instance at the level of the prompt, or at the level of the candidate choice-set.

Without any perturbation, a sufficiently powerful QA system would assign a high confidence score to the correct choice, and low scores to the incorrect ones. This behavior is illustrated on the left side of Fig 1. The question that we aim to investigate in this article is what happens to the confidence distribution of the model after the probes are applied. Theoretically, an 'agnostic' model that is equally indifferent to all incorrect choices would pick an arbitrary choice with uniform probability, leading to a confidence score per choice of $1/n$ where $n$ is the (fixed) *number* of candidate choices that the model has to select from when prompted, and near-zero variance in the confidence distribution. Alternatively, it may well be that the model actually ends up shifting its high confidence in the (previously) correct choice in some non-random, but statistically distinguishable, fashion.

To resolve between these possibilities, we conduct a statistical analysis of the confidence behavior of a fine-tuned, high-performance language model across widely used commonsense multiple-choice QA benchmarks. In the vein of Fig 1, we systematically perturb instances in

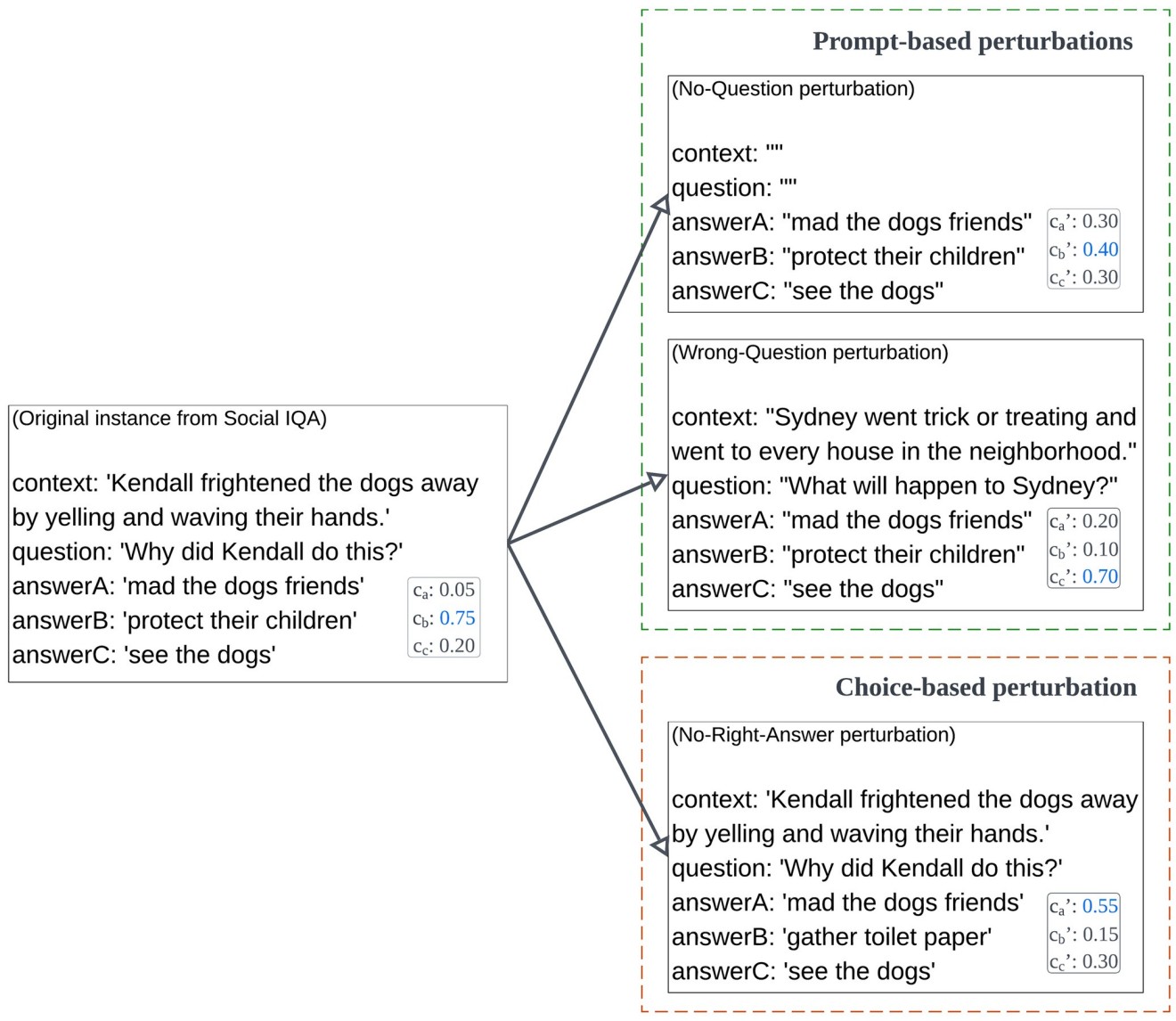

**Fig 1. An example (from the real-world Social Interaction QA benchmark) of the three confusion probes used in this paper as perturbation-based functions.** Two prompt-based perturbations are shown at the top, and a choice-based perturbation is shown at the bottom. In Social IQA, the *prompt* in an *instance* comprises both a context and question, and a set of three candidate choices, of which exactly one is considered correct in the original unperturbed instance. A fine-tuned QA system is able to assign a *confidence score* to every potential choice for a given instance (shown on the bottom right), with confidences across choices summing up to 1. The model will choose the candidate choice with the highest confidence as its predicted correct choice.

these benchmark such that no correct choice exists in response to a given prompt. Because the *manner* in which the question or candidate choices are perturbed might make a difference to the statistical distribution of confidences, we enumerate two related research questions:

1. **Research Question (RQ) 1**: How does the confidence distribution of a sufficiently powerful (i.e., fine-tuned) ROBERTa-based QA system change when the prompt of an instance is modified, such that the original correct choice is no longer correct?

2. **Research Question (RQ) 2**: How does the confidence distribution of a sufficiently powerful (i.e., fine-tuned) ROBERTa-based QA system change when the original correct choice in an

instance is substituted with a new incorrect choice, again rendering the instance without a correct response to the (unmodified) prompt?

Unlike much of the prior work on this subject (further discussed in *Related Work*), we are not seeking to understand the layers of a specific network or how it encodes knowledge, but rather, to understand how these models behave when they are presented with challenging instances without correct solutions (even if such solutions exist). A clear understanding of this behavior, at least in the multiple-choice QA setting, allows us to test whether such language models, which are continuing to be rolled out in commercial products, exhibit statistical behavior that is reasonably predictable when faced with such instances.

## Related work

BERT's original success on discriminative NLP tasks (following a fine-tuning step) has also motivated researchers to adapt it for multi-modal language representation [26, 27], cross-lingual language models [28], and domain-specific language models, including in the legal- [29], finance- [30], patent- [5], medicine- [31–33] and biology-related domains [34]. Due to this widespread use of BERT and the more advanced models based on similar transformer-based networks, it has become important to systematically study the linguistic-cognitive properties of BERT. In prior work, for example, several proposed approaches aimed to study the knowledge encoded within BERT, including fill-in-the-gap probes of MLM [35, 36], analysis of self-attention weights [37, 38], the probing of classifiers with different BERT representations as inputs [39, 40], and a 'CheckList' style approach to systematically evaluate the linguistic capability of a BERT-based model [41]. Evidence from this line of research suggests that BERT encodes a hierarchy of linguistic information, with surface features in the bottom, syntactic features in the middle, and semantic features in the top layers [42]. BERT 'naturally' seems to learn syntactic information from pre-training text without explicit labeling of syntactic structures.

However, it has been found that while information can be recovered from its token representation [43], it does not fully 'understand' naturalistic concepts like negation, and is insensitive to malformed input [35]. The latter is similar to *adversarial* experiments (not dissimilar to adversarial experiments in the computer vision community) that researchers have conducted to test BERT's robustness. Some of these experiments have shown that, even though BERT encodes information about entity types, relations, semantic roles, and proto-roles well, it struggles with the representations of numbers [44] and is also brittle to named entity replacements [45].

The model studied in this paper, RoBERTa [46], is a highly optimized version of the original BERT architecture that was first published in 2019 and improved over BERT on various benchmarks by margins of 0.9 [on the Quora Question Pairs dataset [47]]—16.2 percent [on the Recognizing Textual Entailment dataset [48–51]]. Compared to the original BERT model, RoBERTa implemented several key modifications in its training strategies. First, RoBERTa replaced the original random static masking strategy in BERT's implementation with dynamic masking, a change that offers particular advantages when dealing with larger training datasets or conducting more pertaining steps. Second, BERT used two objectives during pre-training, masked language modeling (MLM) and next sentence prediction (NSP). RoBERTa removed the NSP from the training objective, resulting in a slight improvement in its performance in downstream tasks as compared to BERT. Lastly, the original BERT model was trained with a batch size of 256 sequences, and a character-level byte-pair encoding (BPE) vocabulary of size 30K. RoBERTa's training process involved the use of larger batches (8K sequences) and larger BPE vocabulary (containing 50K subword units). These refinements have further bolstered

RoBERTa's performance, particularly in tasks such as Multi-Genre Natural Language Inference [52] and Question-Based Natural Language Inference [53]. RoBERTa-based models have approached near-human performance on various (subsequently described) commonsense NLP benchmarks.

## Formalism

We begin by defining an *instance I* = (*p*, *A*) as a pair composed of a *prompt p*, and a set *A* = {$a_1, a_2, \ldots, a_n$} of *n* candidate *choices*. The value *n* depends on the benchmark, and is fixed for the benchmark. For example, as shown in Fig 1, *n* = 3 for the Social IQA benchmark. Intuitively, the term *prompt* (rather than *question*) is more appropriate for these problems because the input may not be a proper question. In general, commonsense benchmarks are considered to be natural language *inference* (NLI) benchmarks, which *may* involve QA, but do not have to. Examples of NLI tasks that obey the definition above (and are hence applicable to the study) but are not strictly QA include abductive reasoning and goal satisfaction. Typically, both prompts and choices in these problem domains are represented as statements or paragraphs. We provide examples subsequently when we describe the benchmarks in more detail. In benchmarks like Social IQA, both a statement and question can be part of the prompt, as shown in Fig 1.

Given an instance *I* = (*p*, *A*), we assume that exactly one of the choices $\hat{a} \in A$ is *correct*. Given a language representation model *f* that is designed to handle multi-choice NLI instances (such as the one used in this article), we assume the output of *f*, given *I*, to be the map *C* = {$a_1 = c_1, a_2 = c_2, \ldots, a_n = c_n$} with exactly *n* key-value pairs. Each value in the map is a real value in [0, 1]. We refer to the values in *C* as the *confidence set* that includes the model's confidence $c_i$ per candidate choice $a_i \in A$. The sum of the confidences in the confidence set must add to 1.0. The choice $a' \in A$ that is associated with the highest confidence in *C* is typically considered to have been 'selected' by the model as its response to the originally posed prompt.

We say that *I* is *perturbed* either if *p* is changed in some manner (including being assigned the 'empty string') or if *A* is modified through addition, deletion, substitution, or any other modification of candidate choices. If a perturbation *P* applied on *I* results in the perturbed instance $I_P$ not having any *theoretically correct* choice in response to the prompt, but $\hat{a}$ is still a candidate choice, we refer to $\hat{a}$ as the *pseudo-correct choice*.

An obvious example of when this occurs is a perturbation that 'deleted' the prompt by assigning it the empty string. Since there is no prompt, none of the candidate choices is theoretically correct or incorrect. Assuming that *A* was not modified, the pseudo-correct choice would be $\hat{a}$.

Finally, for the set of incorrect choices $\tilde{A} := A - \{\hat{a}\}$ in an original (unperturbed) instance *I*, we can denote the set of their confidences as $C_{(\tilde{A})}$. Similarly, in a perturbed instance $I_P$, the set of confidences for all incorrect choices except the pseudo-correct choice (which we emphasize is also technically incorrect in a perturbed instance) is denoted as $C_{(\tilde{A},P)}$. Making this terminological distinction will allow us to study the statistical shifting of the confidence distribution between $\tilde{A}$ in the original and perturbed instance, and the change in confidence in the pseudo-correct choice $\hat{a}$ in the perturbed instance compared to the original instance.

It bears emphasizing that $\hat{a}$ is not changed (although its confidence could be significantly changed) when the perturbation is prompt-based, but is necessarily substituted with a different choice when the perturbation is choice-based. The way in which this substitution occurs is subsequently discussed in *Materials and methods*. In contrast, the set $\tilde{A}$, comprising the 'originally incorrect' choices, is always unchanged regardless of the perturbation applied. Furthermore, although we use the three probes that are visually illustrated in Fig 1 in this work, this

formalism would apply to any perturbation function, as long as (i) the perturbation did not occur both at the level of prompt and choices, (ii) resulted in no choice being theoretically correct, and (iii) did not modify the parameter $n$ that is local to the benchmark on which the perturbations are being applied.

## Materials and methods

### Evaluation datasets

The two benchmarks used in the experimental study are described below, with references for further reading. We also provide a representative example in Fig 2. We emphasize that an instance is the combination of the prompt and the candidate choice-set, and only if the prompt is an actual question, should the instance technically be thought of as a QA instance. In the general case, each instance should be thought of broadly as testing natural language inference. In the rest of the discussion, we continue using the proper terms 'instance', 'choice' and 'prompt' (rather than the somewhat inaccurate terms 'QA instance', 'question' and 'answer', respectively) to refer to these concepts.

1. **HellaSwag**: HellaSWAG [54, 55] is a dataset for studying grounded commonsense inference. It consists of 49,947 multiple-choice instances about 'grounded situations' (with 39,905 instances in the training set and 10,042 instances in a held-out set). Each prompt comes from one of two domains–Activity Net or wikiHow–with four candidate choices about what might happen next in the scene. The correct choice is the (real) sentence for the next event; the three incorrect choices are adversarially generated and human-verified, ensuring a non-trivial probability of 'fooling' machines but not (most) humans. Each HellaSwag instance provides two *contexts* as the prompt. UNICORN [56], a model based on the T5 language model, achieves the current highest performance (0.94) of models on this benchmark, which approaches human performance (0.96).

2. **Social IQA**: Social Interaction QA [57, 58] is a QA benchmark for testing social common sense. In contrast with prior benchmarks focusing primarily on physical or taxonomic knowledge, Social IQA is mainly concerned with testing a machine's reasoning capabilities

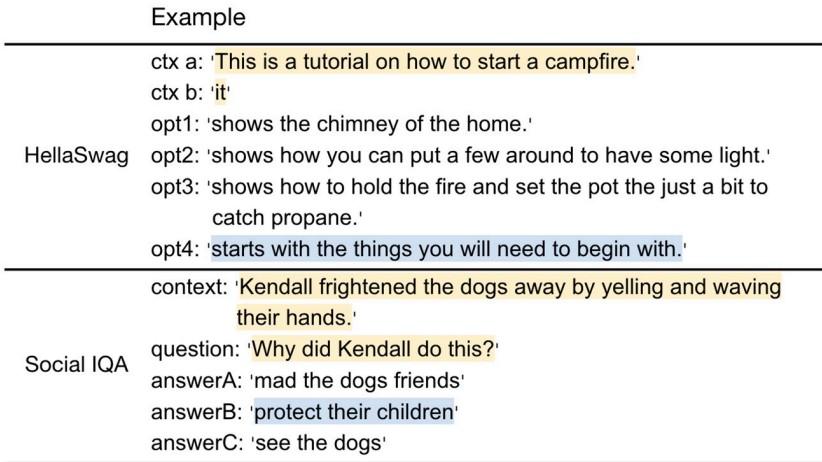

| | Example |
|---|---|
| HellaSwag | ctx a: 'This is a tutorial on how to start a campfire.'<br>ctx b: 'it'<br>opt1: 'shows the chimney of the home.'<br>opt2: 'shows how you can put a few around to have some light.'<br>opt3: 'shows how to hold the fire and set the pot the just a bit to catch propane.'<br>opt4: 'starts with the things you will need to begin with.' |
| Social IQA | context: 'Kendall frightened the dogs away by yelling and waving their hands.'<br>question: 'Why did Kendall do this?'<br>answerA: 'mad the dogs friends'<br>answerB: 'protect their children'<br>answerC: 'see the dogs' |

**Fig 2. Instances (prompt and candidate choice-set) from the two commonsense benchmark datasets used for the experimental study herein.** The prompt is highlighted in yellow, and the correct choice is in blue.

about people's actions and their social implications. Actions in Social IQA span many social situations, and candidate choices comprise of both human-curated answers and 'adversarially-filtered' machine-generated choices. Social IQA contains 33,410 instances in its training set, and 1,954 instances in its held-out set. While Social IQA separates the context from the question, the two collectively constitute the prompt, in keeping with the terminology mentioned earlier. We note that both human- and machine-performance on Social IQA are slightly lower than other benchmarks. Specifically, human accuracy on Social IQA is 0.88, with UNICORN achieving an accuracy of 0.83.

## RoBERTa-based model

RoBERTa is a more optimized re-training of BERT that removes the *Next Sentence Prediction* task from BERT's pre-training, while introducing dynamic masking so that the masked token changes during the training epochs. Larger batch-training sizes were also found to be more useful in the training procedure. Unlike a more recent model like GPT-3 or even ChatGPT, a pre-trained version of RoBERTa is fully available for researchers to use and can be fine-tuned for specific tasks [46].

Unsurprisingly, many of the systems occupying a significant fraction of top leaderboard positions (hosted by the Allen Institute) for the commonsense reasoning benchmarks described earlier are based on RoBERTa (or some other optimized BERT-based model) in some significant manner. All experiments in this paper use a publicly available *RoBERTa Ensemble* model [59], which is a pre-trained RoBERTa-large model and was not developed by any the authors, either in principle or practice, and that can be downloaded and replicated very precisely. It is also worth noting that even recent models that have superseded RoBERTa (such as T5) on the benchmarks are based on transformers as well.

The RoBERTa Ensemble model (henceforth called the *RoBERTa-based Model* for the purposes of this paper) is fine-tuned on each benchmark's respective training set and evaluated on its held-out set to test the model's performance. As mentioned, the fine-tuning of the RoBERTa-based model is conducted using a RoBERTa-large model with specific parameter settings, including a gradient batch accumulation of 3, a maximum of 4 epochs, a learning rate set at 5e-6, 300 warm-up steps, a batch size of 3, and a maximum sequence length of 128. Because there are two benchmarks used in this study, there are two such fine-tuned models. We use the appropriate fine-tuned model when obtaining the confidence distributions of instances (both perturbed and unperturbed) for a given benchmark. Each such trained model was verified to achieve competitive performance over 80% accuracy (on average) over the two benchmarks. Note that the fine-tuned models are not 'aware' of the perturbation, as these are not used during the fine-tuning (in other words, we never apply the perturbation functions to any of the training instances), as our experimental goal is to understand the behavior of the model precisely when it is faced with ambiguous instances that it has not been taught to expect in advance.

## Perturbation methodology

To explore the change in confidence distribution of RoBERTa when the original prompt or candidate choice in an instance is modified in a manner that there is no theoretically correct choice, we designed two prompt-based perturbation functions (*No-Question* probe and *Wrong-Question* probe) for investigating *RQ1*, and one choice-based perturbation function (*No-Right-Answer* probe) for investigating *RQ2*. These functions, named intuitively and

defined below, operate by systematically transforming multiple-choice NLI instances in the two publicly available benchmarks discussed earlier:

The prompt-based perturbation functions that are designed for *RQ1* are:

- **No-Question probe**: This perturbation function modifies an instance by removing the prompt altogether and only retaining the candidate choice-set. For each instance, we replace the prompt with an empty string while still making the RoBERTa-based QA model select a choice from its original choice-set. Note that the RoBERTa-based model is syntactically capable of accepting an empty string as prompt.

- **Wrong-Question probe**: Similar to the No-Question probe, this probe retains the original candidate choice-set for an instance, but 'swaps' the original prompt in an instance with a prompt from another instance (in the same benchmark) that is not relevant to any of the answers. For each instance, we replace the original prompt with a *mismatched* prompt (called the 'pseudo-prompt'). The pseudo-prompt for a given instance is an actual prompt from another randomly selected instance in that benchmark. No change is made to the choice-set. While there is a very small probability that the pseudo-prompt may be correctly answered by a choice from the unmodified choice-set, in practice, we could find no such cases when we randomly sampled and manually checked 25 perturbed instances from each benchmark. As a further robustness check, we conducted another experiment where, instead of randomly sampling an instance from which the pseudo-prompt was selected, we only considered sampling from instances where the prompt did not share any words with the original prompt. The experimental results were not found to change appreciably compared to the simpler replacement protocol described above. Hence, we only report those results for that protocol. Furthermore, to account for randomness, we repeat the experiment five times (per benchmark), and report average performance.

There is precedent for using the *No-Question* perturbation in other natural language tasks. For example, [60] used this probe to test the difficulty of reading comprehension benchmarks. The examples in these benchmarks are tuples consisting of questions, passages, and answers. In their experiments, they analyzed the model's performance on various benchmarks when the test examples were provided with question-only or passage-only information (but not both). Similarly, in an NLI task, [61, 62] re-evaluated high-performing models on hypothesis-only examples. In their experiments, models predicted the label ('entailment', 'neutral' or 'contradict') of a given hypothesis without seeing the premise. The *No-Question* probe in our experiment is similar to the hypothesis-only model, which only provides models with multiple-choice answer-sets but without any associated prompt.

The choice-based perturbation function is designed to explore *RQ2*:

- **No-Right-Answer probe**: This probe retains the prompt and all *incorrect* choices in an instance, but replaces the correct choice with a choice from another instance's choice-set. The model is thus presented with a prompt but no correct choice in response to the prompt (among the presented choices). Note that, in the Wrong-Question probe, the candidate choices are completely unrelated to the corresponding prompt whereas in the No-Right-Answer probe, the non-substituted choices are often found to be semantically related to the prompt. For example, in Fig 1, the original prompt was regarding the Kendall-related context. Following the Wrong-Question intervention, the instance is given a new context and question, which are completely unrelated to the three original Kendall-related answers. In contrast, following the No-Right-Answer perturbation, the context and question remain the same, but the original choice is substituted. Semantically speaking, the substituted choice is less related to the prompt than the non-substituted (but still incorrect) choices. In the

experiments, we substitute each instance's correct choice with a mismatched choice. Similar to the *Wrong-Question* replacement protocol, we randomly sample an instance and substitute the 'correct' choice of the given instance with the correct choice of the randomly sampled instance. We conduct similar robustness checks (and also manual checks) as with the *Wrong-Question* protocol, but again found the results to be similar to that of the described protocol. Hence, only those results are reported. We also account for randomness through five experimental trials.

Earlier, all of these probes were visualized using an actual instance from the Social IQA benchmark in Fig 1. We have publicly made available both the original instances and the perturbed instances, which were obtained by applying the three probes [63]. In the results, we specifically investigate how the confidence distribution of the RoBERTa-based model statistically changes (and whether these changes are consistent across each benchmark) after each perturbation is applied. The confidence of an 'agnostic' model that does not get confused by incorrect choices (even if they are non-trivially related to the prompt or to each other in their surface forms) after perturbation is expected to be randomly distributed, although it can be expected that the language models will prefer the semantically related, non-substituted wrong choices following application of the *No-Right-Answer* probe.

## Metrics

We systematically investigate the change in confidence distribution of RoBERTa (for both research questions) using relatively simple and interpretable metrics. Recall that, in the *Formalism* section, we stated that an unperturbed instance $I = (p, A)$ has exactly one correct choice $\hat{a} \in A$, with the set of its incorrect choices denoted as $\tilde{A} := A - \{\hat{a}\}$. Given $I$, RoBERTa (or a similar such language model) can yield a confidence map $C$ where a value $c_i$ corresponds to the model's confidence for each candidate choice $a_i \in A$. We denote the confidence set of incorrect choices as $C_{(\tilde{A})}$. Generally speaking, for high-performing models like RoBERTa $C_{(\tilde{A})}$ tends to empirically follow a uniform distribution i.e., the model places relatively high confidence (at least on average) on the correct choice, and the rest of its confidence (since all confidence values must add to 1.0) tend to be distributed equally among the incorrect choices.

When $I$ is perturbed by a probe $P$, at either the prompt or the choice level, RoBERTa will generally produce an 'updated' confidence map given the perturbed instance $I_P$. We denote $C_{(\tilde{A},P)}$ as the set which contains the model's updated confidence for the original incorrect answers. Post-perturbation, our expectation of an 'agnostic' model that does not get 'confused' by the prompt or choices is that the confidence of the original incorrect answers $a_i \in \tilde{A}$ should continue to be randomly distributed, even though the absolute values might now be different following the perturbation. We denote such a random confidence distribution of $\tilde{A}$ as $C_{(\tilde{A},random)}$.

By comparing (in aggregate) the confidence distribution of a perturbed instance $C_{(\tilde{A},P)}$ with that of its unperturbed equivalent $C_{(\tilde{A})}$, as well as that of a random distribution $C_{(\tilde{A},random)}$, we can categorize four types of post-perturbation shifts in the distribution. These shifts are discussed below. In all descriptions below, statistical significance (which always occurs between means of two distributions) is determined using a paired t-test test and at the 90% confidence level.

1. **Random-like shift**: This shift is determined to occur if: (i) the mean of $C_{(\tilde{A},P)}$ is found to be significantly different from the mean of the original confidence distribution $C_{(\tilde{A})}$, and (ii) the mean of $C_{(\tilde{A},P)}$ is *not* found to be significantly different from the mean of $C_{(\tilde{A},random)}$.

This kind of shift would be expected from the 'agnostic' system after a perturbation because, even though the confidence distribution for the original incorrect answers has changed, the system still does not exhibit any explicit preference toward any of the original incorrect answers. We note here that these comparison are well-defined because they compare means over the confidences of the same choice-subset (the set of original incorrect choices).

2. **Hybrid shift**: This shift is determined to occur if: (i) no significant difference is found between the mean of $C_{(\tilde{A},P)}$ and the mean of $C_{(\tilde{A},random)}$, and (ii) no significant difference is also found between the mean of $C_{(\tilde{A},P)}$ and the mean of its original distribution $C_{(\tilde{A})}$. This shift suggests that, even if the system has a slight preference for the correct answer in an unperturbed instance, the preference may be almost arbitrary and not statistically significant. When either the prompt or the choice-set is altered and the correct answer is no longer correct, $C_{(\tilde{A},P)}$ tends to be randomly distributed but is still similar to its original distribution $C_{(\tilde{A})}$.

3. **Conservative shift**: This shift is determined to occur if: (i) the mean of $C_{(\tilde{A},P)}$ is found to be significantly different from the mean of random distribution $C_{(\tilde{A},random)}$, and (ii) no significant difference is found between $C_{(\tilde{A},P)}$ and the original confidence distribution $C_{(\tilde{A})}$. Such a change in the confidence distribution suggests that, while the model's confidence may be slightly impacted by the perturbation, its performance remains consistent with that of the original, unperturbed instances. The comparable confidence distribution observed for the same set of incorrect answers even following the perturbation may indicate the presence of a *prior bias*, perhaps learned during either the pre-training phase of the system, or from subtle patterns in the training partition of the benchmark on which it was fine-tuned).

4. **Deviated shift**: This shift is determined to occur if: (i) the mean of $C_{(\tilde{A},P)}$ is found to be significantly different from the mean of the original confidence distribution $C_{(\tilde{A})}$, and (ii) the mean of $C_{(\tilde{A},P)}$ is also found to be significantly different from the mean of the random distribution $C_{(\tilde{A},random)}$. *A priori*, we expect that instances perturbed by the *No-Right-Answer* probe exhibit such behavior because the non-substituted (or original) incorrect answers tend to have higher 'surface' (semantic) similarity to the prompt because of the way in which such benchmarks were originally constructed (with the goal of making the questions more challenging to answer than would be the case if obviously wrong or unrelated choices had been provided as candidates, along with the correct choice).

We keep track of the prevalence of these four types of shifts in instances perturbed by the different probes to empirically investigate both RQs. An additional metric for prompt-based perturbations (RQ1) that we report is the *pseudo-accuracy*, which we define as the fraction of instances in the held-out set where the model ended up selecting the *pseudo-correct* choice (even though it is not correct anymore) to measure the 'performance' of the system on instances perturbed specifically by the prompt-based perturbations. Note that the pseudo-correct choice can only be selected by the model if that choice had the highest confidence among all the choices. Similarly, to further contextualize RQ2, we report additional statistics measurements of RoBERTa's confidence distribution, including the *variance* of confidences in perturbed instances. These measurements are expected to provide additional insights on how the system responds to choice-based perturbations. Such an analysis may lead to more nuanced understanding of the system's behavior when such situations arise.

## Results

### RQ1: Changes in RoBERTa's confidence distribution for prompt-based perturbations

**No-Question probe.**   In Table 1, we report the counts of different shifts in the instances when the *No-Question* probe was applied i.e., the prompt was replaced with an empty string. If the model was equally indifferent between all (incorrect) choices, *random-like* shift counts would dominate the others. In other words, a system that is truly agnostic would sample a choice (at least on average) from the uniform probability distribution over all the available choices when the prompt is the empty string. However, the results show that the random-like shift is rarely observed in either one of the two benchmarks. We find fewer than 200 such instances in both Social IQA and HellaSwag dataset.

We also expect a robust system to have a clear preference for the actual correct answer, in which case there should be a significant difference between the mean confidence of original incorrect choices before and after perturbation. Recall that the confidence of all candidate answers per instance sums to 1. In the original instance (before removing the prompt), the confidence of the actual correct answer yielded by the system should be higher than the reciprocal of the number of candidate choices (which would be expected from a system that is randomly selecting choices, even though there is a correct choice). For the same reason, the mean confidence of the original incorrect choices would then be observed to be *lower* than the reciprocal of the number of candidate choices. If the mean confidence of the incorrect choices following perturbation indeed follows this expectation, it would be significantly different from the mean confidence of these choices prior to perturbation. However, the results in Table 1 show that the model does not fulfill this expectation.

The occurrence of other shifts seems to depend on the benchmark, although the hybrid shift is fairly dominant in both benchmarks. This implies that, when the original incorrect choices are processed by the model after the prompt is removed, the confidence distribution does resemble a random distribution; however, the mean value is not significantly different from the reciprocal of the number of candidate choices. We find 1,473 (out of 1,872) such instances in Social IQA. For HellaSwag, RoBERTa is observed to exhibit similar amounts of *hybrid* (4,828 out of 10,012) and *conservative* (4,749 out of 10,012) shift. The results suggest that RoBERTa is more likely to randomly choose an option for the SocialIQA benchmark after the prompt is removed, compared to HellaSwag.

While the original accuracy of the RoBERTa model used in this paper was reported to be over 80% on both Social IQA and HellaSwag (a result that we also replicated), the pseudo-accuracy data in Table 2 shows that the model may be affected by some type of dataset bias either in the pre-training data, or training partition used for fine-tuning. Clearly, the original 'correct' or pseudo-correct choice is statistically distinguishable even without the prompts. We find no significant difference between the mean confidence of incorrect choices before and

**Table 1. The occurrences (absolute counts / percentage of total instances) of the four different confidence shifts observed in RoBERTa responses when confronted with instances following application of the *No-Question* perturbation.**

|  | Random-like | Hybrid | Conservative | Deviated | Total |
|---|---|---|---|---|---|
| HellaSwag | 181 / 1.8% | 4,828 / 48.2% | 4,749 / 47.4% | 254 / 2.5% | 10,012 / 100% |
| SociaIQA | 147 / 7.9% | 1,473 / 78.7% | 196 / 10.5% | 56 / 3.0% | 1,872 / 100% |

Descriptions of these confidence shifts were described earlier in the *Metrics* section. The number of instances in the last column is the size of the held-out set, because the different shifts form a partition, by definition).

**Table 2. The *pseudo-accuracy* results following each of the two prompt-based perturbations relevant for RQ1.**

| Dataset | Agnostic pseudo-accuracy | No-Question pseudo-accuracy (+/- std. err.) | Wrong-Question pseudo-accuracy (+/- std. err.) | Average confidence difference |
|---------|--------------------------|---------------------------------------------|------------------------------------------------|-------------------------------|
| HellaSwag | 0.25 | 0.66 (+/- 0.004) | 0.61 (+/- 0.005) | 0.060 |
| SocialIQA | 0.33 | 0.46 (+/- 0.011) | 0.40 (+/- 0.011) | 0.022 |

For both prompt-based perturbations, the observed pseudo-accuracy is significantly higher than what should be expected for an agnostic system that is equally indifferent to all incorrect answers following perturbation.

after perturbation for 89.2% Social IQA instances and 95.7% HellaSwag instances. In the absence of the prompts, the confidence distribution of incorrect choices is shown not to be significantly different from that before perturbation. Put together, while the prompts are instructive, when RoBERTa does not see them, it may still yield similar but near-random confidence distributions for the original incorrect choices based on what it learned during pre-training.

The pseudo-accuracy results of RoBERTa (using the 'pseudo-correct' choice as the correct label for accuracy calculations, as discussed earlier) for instances perturbed by the No-Question probe are shown in the third column of Table 2. For reference, we show in Column 2 the 'agnostic' pseudo-accuracy, which by definition, equals the reciprocal of the (benchmark-specific) number of choices per instance as it is equally indifferent to all incorrect answers. The table shows that RoBERTa achieves a high pseudo-accuracy on post-perturbation HellaSwag instances, and closer-to-ideal pseudo-accuracy on Social IQA. The standard error on Social IQA is higher than on HellaSwag, which is a direct consequence of the confidence of pseudo-correct options on Social IQA being more variable (than HellaSwag). Therefore, on HellaSwag the model is clearly more susceptible to dataset bias.

On the other hand, the model does show a marked decrease in pseudo-accuracy on the HellaSwag and Social IQA benchmarks, compared to the original pre-perturbation accuracy. This implies that the prompt plays an important role when present, and the model is using it for answering questions prior to perturbation. However, the results following perturbation also shows that the model does not 'need' the prompt for selecting the originally correct (but now incorrect) answer. This should be an obvious source of concern for common sense evaluations conducting using such benchmarks and models (which is the predominant form of such evaluations in the machine common sense community, as discussed earlier in *Related Work*).

In all cases, the pseudo-accuracy results are significantly different, at a 99 percent confidence level or higher, compared to both the agnostic pseudo-accuracy and the pre-perturbation (or actual) accuracy. The latter is encouraging, but expected, and supports the intuition that the prompt *matters* for the model but the former suggests that it matters less than it should, which future evaluators must bear in mind when interpreting the reasoning capabilities of such models.

**Wrong-Question probe.** We tabulate the changes in RoBERTa's confidence distribution following the application of the *Wrong-Question* probe in Table 3. Consistent with the findings in Table 1, the occurrence of all shifts other than *Hybrid* depends on the benchmark, whereas the hybrid shift is still fairly dominant in both benchmarks, and is observed in more than half of the instances after applying the perturbation. The results suggest that the mean confidence of the incorrect choices for most instances after applying the perturbation is not significantly different from the reciprocal of the (benchmark-specific) number of choices per instance. Although we still observe few occurrences of the agnostic *random-like* shift in both HellaSwag and Social IQA, there are slightly more *random-like* shifts (254 out of 10,012) and fewer *conservative* shifts (3,915 out of 10,012) in HellaSwag, compared to the *No-Question* perturbation.

**Table 3. The occurrences (absolute counts / percentage of total instances) of the four different confidence shifts observed in RoBERTa responses when confronted with instances following application of the Wrong-Question perturbation.**

|          | Random-like | Hybrid | Conservative | Deviated | Total |
|----------|-------------|--------|--------------|----------|-------|
| HellaSwag | 254 / 2.5% | 5,466 / 54.6% | 3,915 / 39.1% | 377 / 3.8% | 10,012 / 100% |
| SociaIQA | 112 / 6.0% | 1,471 / 78.6% | 233 / 12.4% | 56 / 3.0% | 1,872 / 100% |

Descriptions of confidence shifts are provided in the *Metrics* section.

The pseudo-accuracy of RoBERTa on the *Wrong-Question* perturbed instances (in Column 4 in Table 2) is similar to that observed for the *No-Question* perturbed instances (in Column 3), and more aligned to the agnostic performance (in Column 2). Additionally, the average confidence difference between the two probes (Column 5 in Table 2) for each benchmark is not only positive, but also significantly greater than 0 with at least 99 percent confidence. The result implies that the appearance of the wrong prompt introduces more 'doubt' into the model concerning the pseudo-correct choice compared to when the prompt is removed altogether.

We note that the pseudo-accuracy on HellaSwag is still far from the agnostic random performance compared to Social IQA. Therefore, the assumption, commonly made by practitioners using the model, that it needs to be presented with an actually correct option is a powerful one that should not be underestimated. When the options are all incorrect (or sufficiently ambiguous), the model does *not* just randomly or uniformly pick one out as its choice. In other words, there is evidence of dataset bias, which is consistent with the results from the previous section.

### RQ2: Changes in RoBERTa's confidence distribution for the choice-based perturbation

In investigating RQ2, our initial expectation of the changes in RoBERTa's confidence distribution on the *No-Right-Answer*-perturbed instances is similar to that on the prompt-based perturbed instances (RQ1). Specifically, we anticipate a *random-like* shift in RoBERTa's confidence distribution on the perturbed instances, resulting in a nearly uniform probability distribution of original incorrect candidate choices. However, considering the semantic relationship between candidate choices and their corresponding prompt, it appears that the unsubstituted, original incorrect choices were superficially more relevant to the corresponding prompt than the substituted incorrect choice, even though they are all incorrect. Hence, it is possible that the *deviated* shift in RoBERTa's confidence distribution increases following the *No-Right-Answer* perturbation.

Table 4 reports the counts of different shifts in the instances after applying the *No-Right-Answer* probe. The results suggest that RoBERTa exhibited a greater number of random-like shifts when the *No-Right-Answer* probe was applied, compared to the prompt-based

**Table 4. The occurrences (absolute counts / percentage of total instances) of the four different confidence shifts observed in RoBERTa responses when confronted with instances following application of the No-Right-Answer perturbation.**

|          | Random-like | Hybrid | Conservative | Deviated | Total |
|----------|-------------|--------|--------------|----------|-------|
| HellaSwag | 571 / 5.7% | 8,233 / 82.2% | 1,047 / 10.5% | 171 / 1.7% | 10,012 / 100% |
| SociaIQA | 121 / 6.4% | 1,671 / 89.3% | 59 / 3.2% | 21 / 1.1% | 1,872 / 100% |

Descriptions of confidence shifts are described in the *Metrics* section.

**Table 5. A statistical summary of confidence variances observed across candidate choices, on average, before and after the *No-Right-Answer* perturbation was applied.** $\bar{\sigma}_c$ and $\bar{\sigma}'_c$ represent the mean of the variances of confidence in instances before and after perturbation, respectively. The means of $\hat{c}$ and $\hat{c}_{NRA}$ represent the mean confidence of correct (or pseudo-correct) answers yielded by RoBERTa before and after perturbation, respectively.

| Benchmark | $\bar{\sigma}_c$ | $\bar{\sigma}'_c$ | Mean $\hat{c}$ | Mean $\hat{c}_{NRA}$ |
|---|---|---|---|---|
| Hellaswag | 0.36 | 0.31 | 0.78 | 0.20 |
| SocialIQA | 0.33 | 0.30 | 0.69 | 0.17 |

perturbations, but the extent of this increase may vary depending on the benchmark. For example, on Hellaswag, RoBERTa showed twice as many random-like shifts on the *No-Right-Answer* perturbed instances as compared to the *Wrong-Question* perturbed instances; while on Social IQA, there is only a slight 0.4% increase in the counts of random-like shifts. Besides, the hybrid shift remained dominant in both benchmarks. RoBERTa exhibited the hybrid shift on 82.2% of the instances after applying the *No-Right-Answer* perturbation. This also led to a decrease in the frequency of conservative shifts, indicating that the confidence distribution of RoBERTa becomes closer to a random distribution following the perturbation.

In Table 5, we tabulate the confidence variance among all candidate choices, and the mean confidence of correct (or pseudo-correct) answers yielded by RoBERTa before and after applying the *No-Right-Answer* perturbation. If the system were agnostic, we would expect that the mean confidence variances of candidate choices in the original instances will be substantially higher than that in perturbed instances (which would be close to zero if the system had agnostic performance). Meanwhile, the mean confidence of the correct answer before perturbation should be significantly higher than the reciprocal of the number of candidate choices, while the mean confidence of the substituted pseudo-correct answer should be close to or lower than the reciprocal.

Table 5 indicates that the confidence variance (in Column 2) of candidate choices yielded by RoBERTa before perturbation is much higher than zero, implying that it is not equally agnostic to all incorrect choices. Meanwhile, RoBERTa attains close to, or even higher than, 0.7 confidence with respect to the correct answers before perturbation (in Column 4). After applying the *No-Right-Answer* perturbation, RoBERTa is observed to have a similar confidence variance of all candidate choices (in Column 3), while the confidence of the substituted choice (in Column 5) is far below the agnostic performance. The changes in the confidence distribution suggest that the confidence of the correct answer prior to perturbation primarily 'shifts' towards one of the unsubstituted incorrect choices that may have the highest surface semantic similarity to the prompt. Perhaps the most interesting qualitative takeaway from the results is that surface similarity between the prompt and choice can clearly matter a lot, even when the choice is incorrect. For difficult questions or particularly creative (but still correct) answers to such questions, such bias may prove to be problematic.

## Discussion

While the results described in the previous section clearly indicate the non-arbitrary changes induced by the perturbation functions in the RoBERTa-based model's confidence distribution, they do not shed much light on the potential *causes* of these phenomena. For example, is fine-tuning a major driver of the changes that we do observe? In this section, we first provide a discussion using auxiliary analyses to see if syntactic properties or 'irregularities' in the benchmarks themselves may have contributed to the phenomena. However, there may also be other

possible impacts, such as the fine-tuning process of RoBERTa (or other similar transformer-based architectures).

Meanwhile, it remains uncertain whether a system such as RoBERTa can be trained to detect the ambiguity effectively. In other words, it is necessary to determine if the system can already recognize the perturbed instances as 'difficult' or 'ambiguous' cases (to refuse to answer) based on the confidence distribution, or if there is still a need to develop a system to allow the system to do this. Hence, in the second part of the Discussion section, we employ MaxProb [64–66], a popular calibration technique that has been used to investigate the capability of models to detect uncertainty in previous research, to explore whether the changes in confidence distribution induced by the perturbations can be easily detected by the system.

Moreover, while our study has primarily focused on the RoBERTa-based model, it is important to consider the potential generalizability of our findings to other cutting-edge language models, especially high-performing generative models within the GPT family. Despite variations in specific architectural designs and training methodologies between RoBERTa and GPT models, they share fundamental similarities as large-scale transformer-based models. To deepen our insights into the confidence shift phenomenon in transformer-based language models, we also explore, in the latter part of the Discussion section, whether GPT-3.5-Turbo exhibits comparable confidence shifts when exposed to diverse perturbations.

## Analysis of irregularities in benchmarks

As mentioned, it is possible that the specific benchmarks (or the training sets thereof) used to fine-tune the model led to the observed changes in confidence distribution; hence, we begin by conducting an 'analysis of irregularities' whereby we assess the (possibly unusual) prevalence of label imbalance, the distribution of prompt-lengths, and the words-overlap between the prompt and the candidate answers, to include or exclude such phenomena as being indirectly responsible for some of the statistics we observed in the previous section.

Specifically, the label imbalance analysis aims at testing whether the *first* listed choice is labeled in a benchmark's held-out set as correct with a higher-than-random probability. Statistically, we found that the labels tended to be evenly distributed in both benchmarks' held-out sets, and hence, there is no label imbalance or 'choice ordering' bias of any kind. In the three-option Social IQA benchmark, 33.4% and 33.6% of the instances are labeled with 'answerA' and 'answerB' as the correct answer, respectively, which again suggests near-random label distribution of answers A, B and C. Similarly, in the four-option HellaSwag, the frequencies of the four options labeled as the correct answers are 25% (each).

Similarly, we also calculated the average *prompt lengths* (in terms of the number of words) in the two sets comprising the selected and non-selected candidate choices (by the RoBERTa-based system) to determine if there is some kind of a length bias. While we did find that the selected choices tended to be longer than the non-selected choices over all benchmarks, the difference was slight and not statistically significant. For example, in Social IQA, the average length of selected candidate answer is 3.72 words, (with 95% confidence interval of [3.62, 3.82]), while the average length of non-selected candidate answers is 3.69 words. This difference is not significant at the 95% confidence level.

Finally, we analyzed the words-overlap between the candidate choices and the prompt, by grouping all candidate choices into two sets (selected and non-selected), similar to the grouping employed in the analysis above. We found that most of the overlapping words between the prompts and choices (in either set) were *stop-words*. Indeed, when we calculated the Pearson's correlation between the word frequency distributions over the selected and non-selected sets,

the correlation was found to be close to 1.0. Social IQA achieved the lowest correlation value (0.982) between the two sets, while HellaSwag achieved the highest value (0.997).

Taken together, these results suggest that surface irregularities in benchmarks likely cannot serve as an explanation for the experimental results. While a full causal analysis is difficult to conduct experimentally without a larger set of controls (comprising both carefully constructed benchmarks and a broader range of perturbation functions and confusion probes), at least one potential cause could be the *hidden* patterns in the datasets used for fine-tuning. For example, some researchers have found that 'annotation artifacts' [61] can be introduced unintentionally by crowd workers constructing the benchmark (by devising hypotheses and candidate choices). Another hypothetical cause is that models may have picked up the bias during pre-training (e.g., due to increased frequency of some terms). A complete analysis of these hypo-thetical causes for the systematic changes in confidence distribution that we observed is left for future research.

## Can RoBERTa's confidence distribution be used to 'automatically' detect perturbation?

Given the observed results, the question arises as to whether we can semi-automatically detect perturbations and ambiguities in multiple-choice instances (or RoBERTa's response to it) by automatically classifying or 'calibrating' the confidence distribution. Ideally, such a technique would be able to calibrate the original confidence distribution to yield a new distribution that there is more reflective of the model's uncertainty in the responses. There is some precedent for such calibration in the literature. For instance, MaxProb [64–66] has been utilized as an effective calibration technique to detect uncertainty in neural network models. It does so by using the probability assigned by the underlying multiple-choice NLI system to the most likely prediction (i.e., highest probability) among the candidate choices. Previously, MaxProb was found to give good confidence estimates on multiple-choice benchmarks. Here, we use Max-Prob to assess whether the system can distinguish *between* perturbed and unperturbed instances on its own, given the corresponding confidence distribution by RoBERTa. If Max-Prob can make such a distinction, then it opens the door to other possibilities in a full deci-sion-making architecture (e.g., the overall system may abstain from answering if it determines that an instance is perturbed, or resembles a perturbed instance sufficiently strongly).

To train MaxProb, we first (randomly) split the instances in the held-out set in half. We then perturb these instances using one of the three perturbation functions presented earlier, and use both the original and perturbed instances as a 'binary' training set for MaxProb. Spe-cifically, the average MaxProb (as defined above) over this set is treated as a threshold. As in prior usage of the method, when the MaxProb of an instance is higher than the learned thresh-old, the instance is predicted as the original (i.e., non-perturbed) instance. Otherwise, it is con-sidered to be perturbed. The accuracy of MaxProb is defined as the proportion of instances correctly predicted (as perturbed or unperturbed). For evaluation, we use the other half of the held-out set, using the same methodology as we used for the training set (i.e., all instances in the set are first perturbed using the same probe as used for training, followed by 'adding' these perturbed instances back to the original unperturbed set to create an equally balanced mixture of perturbed and unperturbed instances). By construction, the random accuracy is 50 percent.

Table 6 shows both the per-benchmark learned thresholds using the three different confu-sion probes, as well as the corresponding accuracy achieved by MaxProb. Social IQA had the lowest thresholds (among all benchmarks) on all three confusion probes; however, its average learned MaxProb threshold is still 0.7 or higher. Note that RoBERTa should be equally confi-dent about each of its candidate choices on a perturbed instance. Hence, agnostically, the

**Table 6. The learned MaxProb thresholds of different confusion probes on both benchmarks.**

|  | No-Question | Wrong-Question | No-Right-Answer |
|---|---|---|---|
| HellaSwag | 0.82 (0.58) | 0.8 (0.61) | 0.8 (0.62) |
| SocialIQA | 0.70 (0.70) | 0.74 (0.60) | 0.76 (0.55) |

The accuracy of MaxProb to distinguish pre- and post-intervention instances in each evaluation set is shown in brackets. For reference, the random accuracy would be 50% or 0.5 in all cases.

MaxProb on a given benchmark should be the reciprocal of the number of candidate choices (per instance, in that benchmark). Because of the high threshold, MaxProb should help RoBERTa more easily abstain from answering post-intervention instances. However, when we used MaxProb to distinguish perturbed instances in the evaluation set, we found its accuracy to only be slightly higher than random in most cases.

The evidence also suggests that the *No-Right-Answer* instances are the hardest cases for MaxProb, with the *Wrong-Question* instances being the easiest. Hence, not all perturbations are equally difficult for a calibration method. Finally, it bears noting that, while we have exclusively explored the use of MaxProb in these additional experiments, some other selective prediction (or calibration) methods have been proposed more recently [65, 66], and further investigation is warranted as future research to determine the relative effectiveness of these different methods in addressing the ambiguity issue in NLP tasks.

## Additional experiments using GPT-3.5-Turbo

For these additional experiments, we begin by randomly selecting 200 instances from the HellaSwag and SocialIQA held-out sets, respectively. We then applied three proposed perturbations to these instances. For each instance, we presented GPT-3.5-Turbo with both the original unperturbed version and the perturbed version, capturing the model's predictions and the corresponding confidence scores for each candidate answer. Table 7 reports how GPT-3.5 Turbo's confidence changes when exposed to these prompt-based and choice-based perturbations.

One particularly noteworthy finding is GPT-3.5-Turbo's ability to abstain from providing responses to certain perturbed instances. For instance, when faced with No-Question perturbed instances in the HellaSwag dataset, the model declined to provide confidence predictions for 54% of the instances. In other scenarios, it opted not to answer 9.6% of the perturbed instances, on average.

**Table 7. The occurrences (absolute counts / percentage of total instances) of the four different confidence shifts observed in GPT-3.5 Turbo's responses when confronted with instances perturbed by three perturbations.**

|  | *Pertub.* | Random-like | Hybrid | Conservative | Deviated | Total |
|---|---|---|---|---|---|---|
| HellaSwag | NQ | 1 / 0.5% | 84 / 42.0% | 4 / 2.0% | 3 / 1.5% | 92 / 46.0% |
|  | WQ | 2 / 1.0% | 81 / 40.5% | 73 / 36.5% | 35 / 17.5% | 191 / 95.5% |
|  | NRA | 7 / 3.5% | 134 / 67.0% | 30 / 15.0% | 2 / 1.0% | 173 / 86.5% |
| SocialIQA | NQ | 2 / 1% | 45 / 22.5% | 129 / 64.5% | 18 / 9.0% | 194 / 97% |
|  | WQ | 1 / 0.5% | 132 / 66% | 37 / 18.5% | 3 / 1.5% | 173 / 86.5% |
|  | NRA | 0 / 0% | 132 / 66% | 38 / 19.0% | 3 / 1.5% | 173 / 86.5% |

Descriptions of confidence shifts are described in the *Metrics* section. NQ, WQ, and NRA stand for the NO-Question, Wrong-Question, and No-Right-Answer perturbation, respectively.

The confidence shift patterns observed in ChatGPT, as depicted in Table 7, exhibit both similarities and distinctions when compared to the RoBERTa-based model. Across both models, *random-like* shifts were rarely observed in both prompt- and choice-based perturbations. The dominance of the *Hybrid* shift persisted in both benchmarks, regardless of the nature of the perturbation applied. However, GPT-3.5-Turbo exhibited a slightly lower proportion of hybrid shifts compared to RoBERTa. A similar trend was also observed for *Conservative* shifts, which emerged as the second most prevalent shift type but with a slightly less pronounced frequency in GPT-3.5-Turbo. These insights suggest that while factors embedded in model architecture, such as training data and fine-tuning, may influence the observed confidence behaviors, the consistent presence of the confidence shifting phenomenon in high-performing models like GPT-3.5-Turbo remains noteworthy.

## Conclusion

In this article, we proposed to study changes in the confidence distribution induced in a popular BERT-based question answering model by a set of confusion probes. Our methodology and experiments rely on publicly available benchmarks and models, none of which the authors had any role in developing or disseminating, and that can be downloaded and re-used in further experiments. We found evidence that, when instances are perturbed using the prompt-based functions, the confidence distribution of (the originally) incorrect answers in most perturbed instances is close to random, and is similar to the distribution observed before perturbation. The model will still prefer the originally correct (and post-perturbation, 'pseudo-correct') answer even though it is now theoretically incorrect. Hence, it is not agnostic to all incorrect answers, despite the fact that the instances in the held-out set were never seen by it during fine-tuning.

In the case of choice-based perturbations, the model will choose the incorrect choice that, on the surface, seems most closely aligned with the prompt, typically by sharing a greater number of common words, despite its incorrectness. Further analysis, including an analysis of potential 'irregularities' in the benchmarks, suggests that they cannot serve as causal explanations for the observed phenomena. More complex analyses may reveal some correlation between benchmark characteristics and the observations, but a fuller investigation may need detailed linguistic profiling. The results do indicate that the behavior of the model when perturbed is not completely unpredictable. In a real QA-based system, the underlying language model can hypothetically be supplemented with an additional layer or module to discriminate between confidence distributions that arise from clear-cut cases (with one correct answer) and from ambiguous cases (whether intentionally generated or not) where every answer is incorrect, or at best, controversial. Future studies could expand our protocol to consider more probes, benchmarks and models. Such studies may end up providing important insights into the workings and biases of transformer-based models, even as they become larger, more complex and increasingly widespread.

## Author Contributions

**Conceptualization:** Mayank Kejriwal.

**Data curation:** Ke Shen.

**Formal analysis:** Ke Shen.

**Funding acquisition:** Mayank Kejriwal.

**Investigation:** Ke Shen.

**Methodology:** Ke Shen, Mayank Kejriwal.

**Project administration:** Mayank Kejriwal.

**Resources:** Ke Shen.

**Software:** Ke Shen.

**Supervision:** Mayank Kejriwal.

**Validation:** Mayank Kejriwal.

**Visualization:** Ke Shen.

**Writing – original draft:** Ke Shen.

**Writing – review & editing:** Mayank Kejriwal.

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
