## [Decision Letter · Decision Letter 0]

13 Sep 2023

PONE-D-23-07266Quantifying Confidence Shifts in a BERT-based Question Answering System Evaluated on Perturbed InstancesPLOS ONE

Dear Dr. Kejriwal,

Thank you for submitting your manuscript to PLOS ONE. After careful consideration, we feel that it has merit but does not fully meet PLOS ONE’s publication criteria as it currently stands. Therefore, we invite you to submit a revised version of the manuscript that addresses the points raised during the review process.

The authors have proposed investigating the performance of RoBERTa in question-answering, specifically with the use of confusion probes to verify the behavior of such systems when dealing with uncertainty. They show that the model’s confidence is very different from that of an expected model that has no preferences on the incorrect choices when theoretically dealing with scenarios in which there is no correct answer for a question. This suggests potential research paths that deserve further analysis before robust real-world deployment of such systems.

The experimental setup and analysis is sound and the paper is very well-written and detailed. I suggest verifying the minor points raised by the reviewers, in particular addressing the rationale behind the choice of RoBERTa, as well as a discussion on the generality of the findings (e.g., do they stand for LLMs such as Llama, Alpaca, and the GPT family?). Perhaps the inclusion of a small set of experiments in that direction would suffice.

We look forward to receiving your revised manuscript.

Kind regards,

Rodrigo Coelho Barros, Ph.D.

Academic Editor

PLOS ONE

Journal Requirements:

"This work was funded as part of MOWGLI, a project in the DARPA Machine Common Sense (MCS) program, supported by the United States Office Of Naval Research under Contract No. N660011924033."

"MK received funding for this work as a principal investigator of MOWGLI, a project in the Defense Advanced Research Projects Agency (DARPA) Machine Common

Sense program, supported by the United States Office Of Naval Research under

Contract No. N660011924033. The funders had no role in study design, data collection and analysis, decision to publish, or preparation of the manuscript."

Reviewers' comments:

Reviewer's Responses to Questions

**Comments to the Author**

1. Is the manuscript technically sound, and do the data support the conclusions?

Reviewer #1: Yes

Reviewer #2: Yes

2. Has the statistical analysis been performed appropriately and rigorously? 

Reviewer #1: Yes

Reviewer #2: Yes

3. Have the authors made all data underlying the findings in their manuscript fully available?

Reviewer #1: Yes

Reviewer #2: Yes

4. Is the manuscript presented in an intelligible fashion and written in standard English?

Reviewer #1: Yes

Reviewer #2: Yes

5. Review Comments to the Author

Reviewer #1: This paper evaluates the confidence shifts in a ROBERTa model on perturbing QA instances.

#Introduction

In the introduction, the authors explained BERT-like systems and described their choice of ROBERTa. I suggest a paragraph arguing in favor of ROBERTa instead of a "highly optimized version".

They argue that they provide a new set of three confusion probes to test linguistic properties. I suggest removing "linguistic properties" and explaining the three probes later, as the authors did.

The authors provided a significant difference from QA, considering their approach as a prompt, and they presented the Research Questions clearly.

# Related Work

Concerning their related work, it would be interesting to have a ROBERTa discussion in a BERT-based layer of linguistic approaches, the main differences, for example.

#Formalism

The formalism section was interesting in making understanding the probes and correct and incorrect choices clearer.

#Material and methods

They described their material and methods and explained their results regarding their research questions.

#Discussion

They provided a discussion and two analyses: the benchmarks, irregularities, and automatic detections.

#Conclusions

Although they described their conclusion as needing improvement in other experiments' directions, they promoted a substantial analysis concerning the research subject.

Some typos

on on -> on

C(A,P) -> C(Ã,P)

Reviewer #2: This research investigates a BERT-based question-answering model's response to confusion probes, revealing that despite perturbations, the model often maintains a preference for its original correct answer while exhibiting random confidence distributions for incorrect answers, suggesting potential avenues for further analysis and highlighting the need for additional research into transformer-based model behavior and biases.

I find the research questions about what might happen if no correct answer exists for a given prompt among the provided set of choices quite interesting. Moreover, the manuscript is well-written. Motivation and methodology are sound. Therefore, my overall opinion is positive. I have just some comments to help improve the manuscript:

- The paper focuses exclusively on BERT. I missed a discussion about whether the results are generalizable to other types of models of the same nature.

- I do not know if the authors intend to be exhaustive in all the domains in which BERT has application, but some very important ones need to be included, for example, LegalBERT.

- Very little information is given about the RoBERTa model chosen for the experiments. This brings two additional problems: a) it is necessary to go to the literature to understand the details about the training of such a model, and b) it needs to be clarified why such a configuration has been chosen over many other existing ones.

- The following comment is closely related to the previous one. Can the authors foresee analogous behavior/results if the model configuration is changed?

- I do not fully understand the sentence: "incorrect choice that appears superficially closest to the prompt." Does it mean the choice with a smaller semantic distance to the prompt is chosen? How is this defined?

6. PLOS authors have the option to publish the peer review history of their article (what does this mean?). If published, this will include your full peer review and any attached files.

Reviewer #1: No

Reviewer #2: No

---

## [Author Response · Author response to Decision Letter 0]

2 Oct 2023

We thank the editors and all the reviewers for their comments and have attempted to take each concern into account. Among the most significant changes to the revised paper is a more detailed Related Work and Discussion section, and fewer subheadings in other sections. Below, we provide a point-by-point response.

Thank you for stating the following in the Acknowledgments Section of your manuscript: 

"This work was funded as part of MOWGLI, a project in the DARPA Machine Common Sense (MCS) program, supported by the United States Office Of Naval Research under Contract No. N660011924033."

"MK received funding for this work as a principal investigator of MOWGLI, a project in the Defense Advanced Research Projects Agency (DARPA) Machine Common

Sense program, supported by the United States Office Of Naval Research under

Contract No. N660011924033. The funders had no role in study design, data collection and analysis, decision to publish, or preparation of the manuscript."

Response: 

We have removed the funding information from the manuscript and the current funding statement looks fine to us.

Reviewer #1: This paper evaluates the confidence shifts in a ROBERTa model on perturbing QA instances.

#Introduction

In the introduction, the authors explained BERT-like systems and described their choice of ROBERTa. I suggest a paragraph arguing in favor of ROBERTa instead of a "highly optimized version".

Response: 

We have revised the introduction and related work section to offer a more comprehensive explanation of the specific advantages of RoBERTa, highlighting how its modified training techniques and superior performance on NLP benchmarks make it a suitable choice for our research.

They argue that they provide a new set of three confusion probes to test linguistic properties. I suggest removing "linguistic properties" and explaining the three probes later, as the authors did.

Response: 

We have removed “linguistic properties” and rephrased the original sentence “We design and implement a novel set of three confusion probes to test linguistically relevant properties ” to “We design and implement a novel set of three confusion probes to observe how confidence distribution shifts”.

# Related Work

Concerning their related work, it would be interesting to have a ROBERTa discussion in a BERT-based layer of linguistic approaches, the main differences, for example.

Response: 

We have rephrased the RoBERTa paragraph in Related Work section to provide a more detailed explanation of the main differences between RoBERTa and BERT. We believe that this addition enhances the manuscript by offering readers a clearer understanding of the nuanced differences between these two influential models.

Some typos

on on -> on

C(A,P) -> C(Ã,P)

Response: 

We appreciate your careful review of our manuscript and have corrected the mentioned typos. 

Reviewer #2: This research investigates a BERT-based question-answering model's response to confusion probes, revealing that despite perturbations, the model often maintains a preference for its original correct answer while exhibiting random confidence distributions for incorrect answers, suggesting potential avenues for further analysis and highlighting the need for additional research into transformer-based model behavior and biases.I find the research questions about what might happen if no correct answer exists for a given prompt among the provided set of choices quite interesting. Moreover, the manuscript is well-written. Motivation and methodology are sound. Therefore, my overall opinion is positive. I have just some comments to help improve the manuscript:

- The paper focuses exclusively on BERT. I missed a discussion about whether the results are generalizable to other types of models of the same nature.

Response: 

In the revised manuscript, we introduced a new section within the Discussion Section, featuring supplementary experiments employing the GPT-3.5 Turbo model. Given the widespread popularity and exceptional performance of GPT models, we posit that these additional experiments offer robust and representative evidence to support the generalizability of our findings. The results show that while the specific proportions of shifts may exhibit some variance, the general trends in confidence shifts remain remarkably consistent across different models.

- I do not know if the authors intend to be exhaustive in all the domains in which BERT has application, but some very important ones need to be included, for example, LegalBERT.

Response: 

Thanks for the valuable suggestion. We have included more domain-specific applications in the related work section of the revised paper. This includes mention of LegalBERT, ClinicalBERT, FinBERT, and PatentBERT, among others. We think the current Related Work section can provide a more comprehensive overview of the significance of the BERT model in various fields.

- Very little information is given about the RoBERTa model chosen for the experiments. This brings two additional problems: a) it is necessary to go to the literature to understand the details about the training of such a model, and b) it needs to be clarified why such a configuration has been chosen over many other existing ones.

Response: 

We have expanded upon the pre-training and fine-tuning details of RoBERTa in both the Related Work and Materials and Methods sections in the manuscript, aiming to offer readers a clearer understanding of our model choice and setup.

Additionally, regarding the selection of specific fine-tuning parameters, our choice was based on optimizing the model's performance on the validation set. We selected the configuration that yielded the best accuracy results during the fine-tuning process. We hope that these clarifications provide more insight into our model selection and fine-tuning decisions.

- The following comment is closely related to the previous one. Can the authors foresee analogous behavior/results if the model configuration is changed?

Response: 

We appreciate your comment regarding the potential impact of altering the model configuration on our results. While we did not replicate the experiments by varying the RoBERTa model configuration, we did conduct experiments across different models, including ChatGPT (now presented in the Discussion Section), UnifiedQA, and XLNet. Across these diverse models, we observed a certain degree of consistency in the confidence shift patterns, which suggests that the observed behavior is not solely dependent on the specific architecture but is a characteristic of transformer-based language models in general.

Given this observation, we believe that even if we were to modify the configuration of the RoBERTa model, it is likely to exhibit similar behavior in terms of confidence shifts when exposed to perturbations. However, further experimentation would be needed to confirm the extent of this similarity and explore the nuances of different configurations.

- I do not fully understand the sentence: "incorrect choice that appears superficially closest to the prompt." Does it mean the choice with a smaller semantic distance to the prompt is chosen? How is this defined?

Response: 

The phrase 'superficially closest to' does indeed relate to a smaller semantic distance, but it primarily emphasizes a more 'superficial' characteristic, such as the presence of common words shared between the choice and the prompt. To enhance clarity, we have rephrased the original sentence, and the current version reads:

'In the case of choice-based perturbation, the model will choose the incorrect choice that, on the surface, seems most closely aligned with the prompt, typically by sharing a greater number of common words, despite its incorrectness.'

---

## [Decision Letter · Decision Letter 1]

4 Dec 2023

Quantifying Confidence Shifts in a BERT-based Question Answering System Evaluated on Perturbed Instances

PONE-D-23-07266R1

Dear Dr. Kejriwal,

We’re pleased to inform you that your manuscript has been judged scientifically suitable for publication and will be formally accepted for publication once it meets all outstanding technical requirements.

Kind regards,

Rodrigo Coelho Barros, Ph.D.

Academic Editor

PLOS ONE

Reviewers' comments:

Reviewer's Responses to Questions

**Comments to the Author**

1. If the authors have adequately addressed your comments raised in a previous round of review and you feel that this manuscript is now acceptable for publication, you may indicate that here to bypass the “Comments to the Author” section, enter your conflict of interest statement in the “Confidential to Editor” section, and submit your "Accept" recommendation.

Reviewer #1: All comments have been addressed

Reviewer #2: All comments have been addressed

2. Is the manuscript technically sound, and do the data support the conclusions?

Reviewer #1: Yes

Reviewer #2: Yes

3. Has the statistical analysis been performed appropriately and rigorously? 

Reviewer #1: Yes

Reviewer #2: Yes

4. Have the authors made all data underlying the findings in their manuscript fully available?

Reviewer #1: Yes

Reviewer #2: Yes

5. Is the manuscript presented in an intelligible fashion and written in standard English?

Reviewer #1: Yes

Reviewer #2: Yes

6. Review Comments to the Author

Reviewer #1: The authors answered my questions, and they better described their approach. It would be nice to have such a limitation to Roberta in the title instead of a BERT-based but ROBERTA model to delimitate their approach better.

Reviewer #2: The authors have successfully dealt with my comments and suggestions. Therefore, I have decided to upgrade my recommendation.

7. PLOS authors have the option to publish the peer review history of their article (what does this mean?). If published, this will include your full peer review and any attached files.

Reviewer #1: No

Reviewer #2: No

---

## [Editor Report · Acceptance letter]

11 Dec 2023

PONE-D-23-07266R1 

Quantifying Confidence Shifts in a BERT-based Question Answering System Evaluated on Perturbed Instances 

Dear Dr. Kejriwal:

I'm pleased to inform you that your manuscript has been deemed suitable for publication in PLOS ONE. Congratulations! Your manuscript is now with our production department. 

Kind regards, 

on behalf of

Dr. Rodrigo Coelho Barros 

Academic Editor

PLOS ONE